# Aryl Hydrocarbon Receptor and Dioxin-Related Health Hazards—Lessons from Yusho

**DOI:** 10.3390/ijms22020708

**Published:** 2021-01-12

**Authors:** Masutaka Furue, Yuji Ishii, Kiyomi Tsukimori, Gaku Tsuji

**Affiliations:** 1Research and Clinical Center for Yusho and Dioxin, Kyushu University Hospital, Fukuoka 812-8582, Japan; gakku@dermatol.med.kyushu-u.ac.jp; 2Department of Dermatology, Graduate School of Medical Sciences, Kyushu University, Fukuoka 812-8582, Japan; 3Division of Pharmaceutical Cell Biology, Graduate School of Pharmaceutical Sciences, Kyushu University, Fukuoka 812-8582, Japan; ishii@phar.kyushu-u.ac.jp; 4Department of Obstetrics, Perinatal Center, Fukuoka Children’s Hospital, Fukuoka 813-0017, Japan; tsukimori.k@fcho.jp

**Keywords:** Yusho, aryl hydrocarbon receptor, chloracne, dioxin, 2,3,4,7,8-pentachlorodibenzofuran, polychlorinated biphenyl, reactive oxygen species, nuclear factor-erythroid 2-related factor-2, treatment

## Abstract

Poisoning by high concentrations of dioxin and its related compounds manifests variable toxic symptoms such as general malaise, chloracne, hyperpigmentation, sputum and cough, paresthesia or numbness of the extremities, hypertriglyceridemia, perinatal abnormalities, and elevated risks of cancer-related mortality. Such health hazards are observed in patients with Yusho (oil disease in Japanese) who had consumed rice bran oil highly contaminated with 2,3,4,7,8-pentachlorodibenzofuran, polychlorinated biphenyls, and polychlorinated quaterphenyls in 1968. The blood concentrations of these congeners in patients with Yusho remain extremely elevated 50 years after onset. Dioxins exert their toxicity via aryl hydrocarbon receptor (AHR) through the generation of reactive oxygen species (ROS). In this review article, we discuss the pathogenic implication of AHR in dioxin-induced health hazards. We also mention the potential therapeutic use of herbal drugs targeting AHR and ROS in patients with Yusho.

## 1. Introduction

In 1968, more than 2000 people living in Western Japan were intoxicated by high concentrations of polychlorinated dibenzofurans (PCDFs), polychlorinated quaterphenyls, polychlorinated dibenzodioxins (PCDDs), and polychlorinated biphenyls (PCBs), which had contaminated edible rice bran oil [1,2,3]. Considering the toxic equivalency factor (TEF) and toxic equivalent (TEQ) of each congener [4,5], PCDFs and especially 2,3,4,7,8-pentachlorodibenzofuran (2,3,4,7,8-PeCDF) were identified as the most likely causative compounds of toxicity [6]. PCDFs and 2,3,4,7,8-PeCDF comprised 77 and 58% of the total TEQs, respectively [6]. The TEF of 2,3,4,7,8-PeCDF is 0.3 compared with the most toxic component 2,3,7,8-tetrachlorodibenzo-*p*-dioxin (2,3,7,8-TCDD, TEF = 1) [5]. The contaminated rice bran oil contained extremely high concentrations of PCBs. However, the toxicity of PCBs was low because of their low TEQs [4,5]. The measurement of blood concentrations of PCBs was simple immediately after onset, but it was difficult to measure the blood levels of 2,3,4,7,8-PeCDF with sufficient sensitivity, reliability, and reproducibility. However, technical improvement allowed us to do this in 2003 [7].

The intoxicated victims exhibited various symptoms including general malaise, headache, abdominal pain, paresthesia/numbness/pain of the extremities, joint swelling and arthralgia, cough and sputum, and Meibomian gland hypersecretion [8,9]. Among them, most characteristic features were chloracne and hyperpigmentation [8,9,10,11,12]. Laboratory examination revealed an elevation of serum triglyceride levels [13]. These disorders are collectively called Yusho (“oil disease” in Japanese). The early clinical and basic research results were published in the English-language book “Yusho” and the Supplementary Issue “Long-term effects of polychlorinated biphenyls and dioxins in humans—Lessons from Yusho” in the *Journal of Dermatological Science*, both of which are freely available for download [14]. Review articles on Yusho are also available in the literature [15,16,17,18,19]. The purpose of this review article was to highlight dioxin-related health hazards by focusing and aryl hydrocarbon receptor (AHR) function and to propose therapeutic strategies.

## 2. Methods

Using free text and key words, we searched PubMed (https://www.ncbi.nlm.nih.gov/pubmed/advanced), Google Scholar (https://scholar.google.co.jp/schhp?hl=ja) and Japan Medical Abstracts Society (https://search.jamas.or.jp/) databases. Studies were limited to those published from January 1968 to November 2020 with no language restrictions. We included studies of Yusho, Yucheng, Seveso, dioxin, PCB, AHR, reactive oxygen species (ROS), chloracne, pigmentation, lung, cough, sputum, nerve, paresthesia, triglyceride, sleep, circadian clock, immune response, Th22, mortality, carcinogenesis, perinatal, infant, offspring, breast milk, sex ratio, blood concentration, half-life, NRF2, treatment, and/or phytochemicals.

## 3. AHR and Its Biological Function

AHR is a transcription factor and receptor for small-molecule chemicals including dioxins, the environmental pollutant benzo(a)pyrene, phytochemicals such as flavonoids, tryptophan photoproducts, bioproducts of cutaneous and intestinal microbiomes, and drugs [20,21,22]. In a nonstimulated condition, AHR forms a protein complex with heat shock protein 90, hepatitis B virus X-associated protein 2, p23, and c-Src in the cytoplasm [20,23,24,25,26]. Upon ligation, AHR exits the chaperon protein complex, translocates to the nucleus, heterodimerizes with AHR nuclear translocator (ARNT), binds xenobiotic response elements, and regulates the expression of target genes [20,23,24,25,26]. Among numerous target genes [27], cytochrome P450 1A1 (CYP1A1), CYP1A2, and CYP1B1 are xenobiotic-metabolizing enzymes specific for AHR activation, and these enzymes are known to metabolize AHR ligands [20,23,24,25,26,27,28]. ROS are generated during the metabolic process by CYP1A1 and CYP1B1 (Figure 1) [20,24]. Some AHR-binding pollutants also exert endocrine-disrupting activity [26].

One of the well-examined endogenous ligands is tryptophan photoproduct 6-formylindolo[3,2-b]carbazole (FICZ) which is induced by ultraviolet ray irradiation [29]. FICZ is a high-affinity ligand for AHR [29]. Although FICZ and flavonoids are rapidly metabolized by CYP1A1, biologically stable TCDD and dioxins are not easily metabolized, and large amounts of ROS are continuously produced [20,28,29,30,31,32]. In fact, biomarkers for oxidative stress are elevated in the blood and urine of patients with Yusho [33,34]. Excessive ROS damage DNA, proteins, and lipids and induce the production of anti-inflammatory cytokines [24,32,35,36]. As the production of ROS by TCDD is canceled in AHR- or CYP1A1-depleted cells [37], the AHR–CYP1A1 axis is critical for dioxin–AHR-mediated ROS generation. It is known that the gene expression of AHR is influenced by polymorphism of the nuclear transcription factor NF1C [38].

Cells are equipped with an antioxidative system to neutralize oxidative stress to ensure their survival. Ligation of AHR by TCDD induces the production of ROS, which in turn activate nuclear factor-erythroid 2-related factor-2 (NRF2), the master transcription factor for antioxidative enzymes such as NAD(P)H quinone oxidoreductase 1 [39,40]. However, long-lasting ROS generation in response to stable TCDD is likely to overwhelm the NRF2-mediated antioxidative capacity, and cells ultimately display oxidative stress [24,32].

AHR is expressed in almost all cells including epithelial, mesenchymal, and hematopoietic cells [20,24,26,28,32]. As the toxic responses by TCDD are not detected in AHR-deficient mice, AHR is primarily involved in the toxicity of TCDD in all cells [41,42]. Therefore, the human health hazards by dioxins are also likely attributable to excessive ARH activation and prolonged oxidative stress induced by dioxins.

## 4. AHR and Clinical Symptoms in Yusho

### 4.1. Chloracne and Hyperpigmentation

Chloracne and hyperpigmentation are typical skin symptoms in patients exposed to high concentrations of PCBs and dioxins. Chloracne and hyperpigmentation were observed in patients with Yusho [10], patients with Yucheng exposed to PCBs and PCDFs in Taiwan in 1976 [43], and victims who were exposed to TCDDs following the explosion of a pesticide-manufacturing plant in the Seveso disaster in 1979 [44,45,46]. Blood levels of PCBs and PCDFs in patients with Yusho are significantly higher than those in normal controls even more than 50 years since onset, and their levels are positively correlated with the severity of chloracne and hyperpigmentation [10].

Histopathologically, early lesions of chloracne start as hyperkeratosis and keratotic plug formation in the orifices of follicles and sebaceous glands. Sebum production by sebocytes in the sebaceous glands gradually decreases with keratinocytic metaplasia of sebocytes, leading to the complete loss of sebocytes. The folliculosebaceous unit finally forms a cystic appearance [47]. Recent studies clearly revealed a pathogenic role of AHR in the development of chloracne and hyperpigmentation [48,49].

Human epidermis is composed of stratified layers; basal, spinous, granular, and cornified layers. Basal keratinocytes divide, leave the basal layer, and move through the spinous layer to the granular and cornified layers. The differentiated cornified cells and their intercellular lipids maintain the skin barrier function. In this terminal differentiation process, keratinocytes coordinately express various barrier-related proteins such as filaggrin, loricrin, and involucrin in the granular layer and differentiate into cornified cells by forming cornified cell envelopes. AHR signaling accelerates epidermal terminal differentiation by upregulating the expression of these barrier-related proteins [50,51,52,53,54,55,56,57,58]. Dioxins are known to accelerate epidermal terminal differentiation via excessive AHR activation (Figure 2) [59,60,61].

Sebaceous glands open to hair follicles, and they are composed of sebocytes. Sebocytes are special keratinocytes that produce sebum following differentiation. Sebocytes also abundantly express AHR [62]. Notably, excessive activation of AHR by TCDD inhibits sebum production by sebocytes and induces sebocyte-to-keratinocyte metaplasia [62] (Figure 2). Keratinocytic metaplasia of sebocytes induced by TCDD leads to a loss of sebocytes and sebaceous glands, retention of keratinous materials in hair follicles, formation keratinous plug formation in the follicular orifice, and finally follicular cyst formation, which is called chloracne (Figure 2) [62]. The TCDD-mediated acceleration of epidermal terminal differentiation appears to be AHR-dependent because it is cancelled by the blockade of AHR signaling [63].

In physiological conditions, AHR participates in the homeostatic regulation of skin barrier function by coordinately upregulating barrier-related proteins [48,49]. However, excessive activation of AHR by dioxins causes chloracne via keratinocytic metaplasia of sebocytes and follicular cyst formation [62]. In addition, excessive AHR activation induces proinflammatory cytokine production, which accelerates the recruitment of inflammatory cell infiltration around the folliculosebaceous units [35,36]. This is the pathogenesis of chloracne. Similar keratinocytic metaplasia has been observed in the palpebral Meibomian glands of monkeys treated with dioxins [64].

Interleukin (IL)-24 is a keratinocyte-derived cytokine that suppresses epidermal terminal differentiation [65]. Notably, *IL24* is a downstream target gene of AHR signaling, and its gene expression is increased by AHR ligation [66]. Polymorphism of *AHR* is known to affect *IL24* expression [66]. Thus, it is plausible that AHR signaling ultimately results in the acceleration of epidermal terminal differentiation despite the simultaneous activation of an IL-24–mediated negative feedback mechanism [67].

AHR is also expressed in melanocytes [49,68]. Activation of AHR by dioxins is known to upregulate the production of melanin pigments in melanocytes, and it manifests as clinical hyperpigmentation [49,68,69]. Cutaneous hyperpigmentation is actually correlated with the blood levels of PCB and 2,3,4,7,8-PeCDF in patients with Yusho [10,12,70,71,72,73]. Cutaneous hyperpigmentation was observed in fetuses delivered by mothers with Yusho who were exposed to high concentrations of 2,3,4,7,8-PeCDF [74]. Tobacco smoke contains various chemical pollutants with AHR ligand activity, and it increases melanogenesis in human melanocytes [75]. Ultraviolet B (UVB) irradiation synergistically enhances tobacco smoke-induced melanogenesis [76]. It is known that UVB irradiation converts tryptophan to FICZ, which is a high-affinity ligand of AHR [29,36]. A recent epidemiological study assessing the relationship between particulate matter 2.5 (PM2.5) and skin eruptions revealed that individuals with high exposure to PM2.5 exhibit a higher frequency of senile pigmented freckles [77]. Because PM2.5 also contains a number of environmental pollutants with AHR ligand activity [68,69], activation of AHR by PM2.5 may be potentially involved in the occurrence of hyperpigmentation.

### 4.2. Cough and Sputum

Cough and sputum are common symptoms found in patients with Yusho, and their occurrence was significantly correlated with the blood levels of PCB and 2,3,4,7,8-PeCDF [71,73]. AHR is present in tracheal epithelial cells. Once activated by dioxins, AHR signaling generates ROS and subsequently increases the production of MUC5AC (mucin product), which may cause cough and sputum (Figure 3) [78,79,80]. Workers in a car manufacturing factory (N = 201) displayed worse pulmonary function than people living near the factory (N = 222) [81]. The forced vital capacity and forced expiratory volume in 1 s were negatively correlated with the cumulative exposure to PCDDs and PCDFs in workers [81]. As chronic obstructive pulmonary disease (COPD) is associated with smoking habits, AHR activation by tobacco smoke dioxins may be potentially involved in the pathogenesis of COPD [82].

### 4.3. Paresthesia, Pain, and Numbness of the Extremities

Patients with Yusho continue to experience paresthesia, pain, and numbness of the extremities, which are significantly associated with the blood levels of PCB and 2,3,4,7,8-PeCDF [70,73]. When rats are systemically treated with benzo(a)pyrene, the nerve conduction velocity of Aβ fibers, but not Aγ and C fibers, is decreased in the dorsal root, and synaptic transmission in substantia gelatinosa neurons of the spinal dorsal horn is suppressed (Figure 3) [83]. The decrease in conduction velocity may potentially reflect the paresthesia and numbness of the extremities. The diameter of Aβ fibers is larger than those of Aγ and C fibers; therefore, it is known that Aβ fibers are more sensitive to mechanical damage than Aγ and C fibers [84]. Although the pathogenic mechanism of benzo(a)pyrene-mediated decreases of the conduction velocity in Aβ fibers remains unclear, demyelination of Aβ fibers is suspected [83]. Although the myelin gene is not the direct target of AHR signaling, AHR enhances β-catenin levels and stimulates the binding of β-catenin to myelin gene promoters in Schwann cells [85]. Another study revealed that the TCDD–AHR axis enhances the production of neurofilaments involved in the differentiation and degeneration of nerve fibers [86].

### 4.4. Blood Triglyceride Levels

Hyperlipidemia, especially hypertriglyceridemia, is one of the characteristic laboratory abnormalities in patients with Yusho [87,88]. Serum triglyceride levels are positively correlated with blood PCB levels in patients with Yusho [87,88]. Hepatocytes abundantly express AHR [89]. The production of lipid droplets is significantly increased in TCDD-treated hepatocytes, and this effect is dependent on AHR-mediated CYP1A1 induction [89]. By contrast, serum triglyceride levels are decreased in AHR knockout mice [90]. Although the mechanisms of AHR-mediated hypertriglyceridemia remain unclear, a possible explanation is the increased degradation of 17β-estradiol (E2) in hepatocytes [91] (Figure 4). E2 downregulates sterol regulatory element-binding protein-1c and inhibits the neogenesis of triglycerides [91,92]. E2 also enhances β-oxidation of fatty acids by increasing the production of peroxisome proliferator-activated receptor-α [91,92]. Activation of AHR upregulates the expression of CYP1A1, which efficiently degrades E2. Decreased E2 levels result in triglyceride neogenesis [91]. Nonalcoholic fatty liver disease (NAFLD) has recently attracted attention because it promotes the development of nonalcoholic steatohepatitis, liver cirrhosis, and hepatoma [93]. Several researchers suggested that the pathogenesis of NAFLD is similar to the abnormal lipid metabolism induced by AHR activation [91,94].

### 4.5. Sleep Disturbance

The circadian clock regulates various aspects of body function including sleep/awake rhythms, body temperature, and endocrine system functioning [95,96,97]. Although the regulatory mechanisms of circadian rhythms are complex, the mutually competitive regulation of several key molecules plays pivotal roles [95,96,97]. The center of the circadian clock is located in the suprachiasmatic nucleus (SCN) in the brain [95,96,97]. In SCN neurons, heterodimeric BMAL and CLOCK proteins upregulate the production of PER and CRY proteins, which form a heterodimer and inhibit the expression of the BMAL–CLOCK heterodimer to generate a periodic circadian rhythm [95,96,97]. AHR is also present in SCN neurons. Lipophilic dioxins can permeate the blood–brain and activate AHR in SCN neurons [97,98]. Notably, AHR can dimerize with BMAL and suppress the BMAL–CLOCK heterodimer, thereby potentially disrupting circadian rhythms via BMAL–CLOCK and PER–CRY interactions [97,98]. A questionnaire and face-to-face survey conducted in 140 patients with Yusho revealed that moderate to severe sleep disturbance was recognized in 51.8% of patients, whereas restless leg syndrome was reported by 30.7% of patients [99]. Their blood levels of 2,3,4,7,8-PeCDF were significantly associated with reduced quality of sleep and a high prevalence of restless leg syndrome [99].

### 4.6. Arthralgia and Brusitis

Patients with Yusho or Yucheng frequently present with arthralgia/arthritis and bursitis [9,71,73,100]. Although the pathogenic role of AHR activation in joint disorders is not fully understood, TCDD-mediated AHR activation induces the apoptosis of chondrocytes via ROS generation [101,102]. Physical function had deteriorated, especially in male patients with Yusho [103]; however, osteoporosis was not evident in patients with Yusho as assessed using reduced bone mineral density [104].

### 4.7. Immune Response

AHR activation is known to shift immune responses toward T helper 17 (Th17), Th22, and T regulatory (Treg) cell differentiation [28,105]. AHR is not expressed in Th1 and Th2 cells, but it is strongly expressed by Th17/Th22 cells and weakly expressed by Treg cells in mice [106]. In AHR-deficient mice, the differentiation of Th1 and Th2 cells is not affected, whereas IL-22 production is specifically downregulated in Th17 cells [106]. Transient activation of AHR promotes Th17 cell differentiation and upregulates IL-22 production [107]. Conversely, sustained activation of AHR by TCDD increases the number of Treg cells [108]. These results suggest that AHR may differentially activate immune responses in a ligand-dependent manner [28,107,108].

The number of natural killer (NK) cells is increased and the serum levels of cytotoxic T lymphocyte-associated antigen-4, a marker of activated NK cells, are also elevated and positively correlated with blood PCDF levels in patients with Yusho, whereas the number of Treg cells is not altered [109]. Serum levels of IL-17, IL-23, IL-1β, and tumor necrosis factor-α are also elevated in patients with Yusho, suggesting an ongoing inflammatory process induced by dioxins in these patients [110]. Contrarily, serum levels of chemokine (C-C motif) ligand 5 (CCL5) are significantly lower in patients with Yusho than in normal healthy controls [111]. In parallel, AHR activation significantly reduces CCL5 production in keratinocytes treated with benzo(a)pyrene [111]. The physiological implication of AHR-mediated CCL5 inhibition warrants further research.

Murine studies revealed that exaggerated AHR activation by dioxins can potentially aggravate atopic dermatitis [112,113]. However, the lifetime prevalence of atopic dermatitis in patients with Yusho was 8.8% (56/638), which was lower than that in the general population in Japan [114,115]. In the Seveso cohort, children born to mothers exposed to TCDD exhibited a lower prevalence of atopic dermatitis, and the risk of atopic dermatitis was negatively correlated with the speculated TCDD exposure levels in mothers [116]. In this study, the prevalence of asthma or rhinitis was also comparable between the children of TCDD-exposed women and their normal healthy counterparts [116]. These studies intimate that humans are more tolerant of the allergy-inducing effects of dioxins than mice.

### 4.8. Mortality Rate and Carcinogenesis

The toxicity of dioxins is diverse among animal species. In general, humans are more tolerant to the toxic effects of dioxins than mice and rats [117,118]. Tenfold higher TCDD concentrations are necessary to activate human AHR than needed to similarly activate murine AHR [117,118]. Even in mice, strain differences in dioxin sensitivity cannot be neglected [117,118]. Hamsters are roughly 1000-fold more tolerant to dioxins than guinea pigs [117,118]. The diversity of dioxin toxicity is attributable to interspecies differences in the affinity of dioxins for AHR for dioxins as well as target gene differences after AHR activation [118,119]. Therefore, both animal models and human epidemiological studies are indispensable for assessing the long-term health effects of dioxins.

Pesticides contaminated with TCDD (Agent Orange) were spread during the Vietnam War in 1961 to 1971. Mortality surveillance conducted in 1992–2005 for Korean Vietnam veterans revealed that the all-cancer mortality rate was associated with Agent Orange exposure [120,121]. Concerning individual cancers, Agent Orange exposure was associated with the mortality of stomach, intestinal, liver, laryngeal, lung, bladder, and thyroid cancers as well as chronic myelogenic leukemia [120,121]. In addition, mortality associated with myocardial infarction, COPD, and liver cirrhosis was also related to Agent Orange exposure [121]. 

A mortality study of patients with Yusho was conducted in 2007, 40 years after the 1968 incident, and it revealed that the mortality rates for all cancers, lung cancer, and liver cancer were significantly increased in male patients with Yusho compared with that in the general male population [122]. A similar epidemiological study was conducted in patients with Yucheng in Taiwan in 2008, 30 years after the 1979 incident, and the results illustrated that the mortality rates associated with all cancers, cardiovascular diseases, and musculoconnective tissue diseases were significantly higher than those of the general population [123]. A meta-analysis of Yusho and Yucheng studies revealed that the morality rates attributable to all cancers, lung cancer, cardiac diseases, and liver diseases were increased in males, whereas that associated with liver cancer was elevated in females [124]. Updated Yusho cohort data on the standardized mortality ratio (SMR) for the period of 1968–2017 revealed that increased mortality was observed for all cancers (SMR = 1.22, 95% confidence interval [CI] = 1.02–1.45) and lung cancer (SMR = 1.59, 95% CI = 1.12–2.19) among males, and increased mortality was observed for liver cancer among females (SMR = 2.05, 95% CI = 1.02–3.67) [125]. By contrast, relative and net survival were not affected by dioxin exposure in patients with Yusho [126,127]

There are controversial results illustrating that AHR activation accelerates or inhibits carcinogenesis and tumor promotion [128,129,130,131]. The blood concentrations of 2,3,4,7,8-PeCDF are higher in female patients with Yusho than in male patients [10,132]. Nevertheless, the long-term effects of dioxins in patients with Yusho and Yucheng indicate that dioxin exposure likely increases cancer-related mortality more strongly in men than in women, suggesting that men may be more susceptible to dioxins than women in terms of oncogenesis.

### 4.9. Perinatal Abnormalities and the Impact in Offspring

AHR is abundantly expressed in the placenta to form a placental barrier between the mother and fetus [133]. Dioxins pass through the placenta and reach the fetus [134,135,136]. The transplacental transfer rate of dioxin compounds differs for each congener. It has been reported that the transplacental transfer rate of highly toxic 2,3,4,7,8-PeCDF (TEF = 0.3) was low, whereas that of the low-toxicity compound 1,2,3,4,6,7,8-HeptaCDD (TEF = 0.01) was high [134]. Newborn babies delivered by mothers with Yusho occasionally exhibited hyperpigmentation, called fetal Yusho. Children with fetal Yusho and their mothers had significantly higher blood concentrations of 1,2,3,6,7,8-HexaCDD (TEF = 0.1) than their counterparts without hyperpigmentation [135]. An analysis of the dioxin distribution in maternal blood and fetal tissues illustrated that compared with the levels in maternal blood (arbitrarily set at 100%), the comparative TEQs were 81.5% in the placenta, 37.2% in cord blood, 53.2% in the vernix caseosa, 18.2% in the meconium, and 9.2% in amniotic fluid [136]. These results indicate that dioxins are excreted through the vernix caseosa and meconium in fetuses.

Dioxins in mothers can be transferred to their children via breast milk. The concentrations of 2,3,4,7,8-PeCDF in breast milk were extremely high at 309.6 ± 71.2 and 745.9 ± 52.9 pg/g-lipids, respectively, in two mothers with Yusho, compared with 8.5 ± 3.1 pg/g-lipids in normal healthy mothers (N = 9) in 1988–1991 [137]. These results indicate that breast milk feeding should be avoided in mothers with high dioxin exposure.

The placental expression of AHR highlights the potential risks of perinatal events in mothers with Yusho related to high dioxin poisoning. Perinatal events in women with Yusho who became pregnant in 1968–1977 (within a decade after onset) were compared with those in women who became pregnant in 1958–1967, 1978–1987, and 1988–1997. Induced abortion and preterm delivery rates were significantly higher in 1968–1977 than in 1958–1967 (10 years before Yusho onset) [138]. Spontaneous abortion and pregnancy loss were also more frequent in 1968–1977 than in 1958–1967, albeit without significance [138]. In addition, maternal exposure to higher concentrations of dioxins significantly decreased the birth weight of newborns [139]. In a Seveso cohort with air-born exposure to TCDD, no association was found between estimated TCDD levels in mothers and spontaneous abortion, fetal growth, gestational length, or neurocognitive/physical function in children [140,141]. Regarding pregnancy outcomes, increased stillbirth rates were reported in a Yucheng cohort, and increased time to pregnancy and infertility were observed in a Seveso cohort [142,143,144].

A decreased sex ratio (male/female) has been reported in the F1 and F2 generations of mothers and fathers exposed to high concentrations of PCBs and dioxins in mice and humans [145,146]. In a Seveso cohort, a decreased sex ratio was reported [147]. Conversely, a decreased sex ratio was not initially observed in a Taiwanese Yucheng cohort [148], but a subsequent analysis revealed a decreased sex ratio among children born to fathers exposed to PCBs and dioxins before 20 years of age [149].

Experimental results in animal models remain inconclusive [150]. A recent murine study revealed that the sex ratio was decreased in the F1 generation sired by TCDD-exposed male mice, and this effect was not observed in AHR-deficient male mice [151]. AHR protein localizes to spermatids and Leydig cells in the testes, and TCDD treatment does not alter the number of X or Y chromosome-bearing sperm [151]. These results suggest that TCDD differentially affects the functional aspects of X- or Y-chromosome-bearing sperm [151]. In a Yusho cohort, the sex ratio was not decreased in the F1 and F2 generations. However, the sex ratio in the F1 generation born to mothers exposed to PCBs and dioxins before the age of 20 years tended to be lower (P = 0.06), and it was significantly decreased in the F2 generation born to F1 mothers (P = 0.02) [152]. Multigenerational and transgenerational effects of dioxins were recently reviewed elsewhere [153].

In a Seveso cohort, the ovarian function of mothers was not associated with TCDD exposure [154], but the number and motility function of sperm in breastfed male children born to mothers exposed to TCDD were significantly lower than those in their unexposed counterparts [155]. Male patients with Yucheng also had abnormalities in their sperm such as morphological changes, decreased counts, and reduced capability for oocyte penetration [156,157]. The cholesterologenic and steroidogenic properties of testes were reduced in fetuses born to mothers treated with TCDD (TEF = 1) compared with the findings in children born to women treated with 2,3,7,8-tetrachlorodibenzofuran (TEF = 0.1) [158]. These findings support the notion that high dioxin exposure can affect sperm in males. 

The Seveso Women’s Health Study revealed that neurocognitive and physical functions such as walking speed, coin flipping, sitting reach, grip strength, digit span, and spatial span were not associated with the estimated TCDD exposure levels in women [159]. In the Seveso Second Generation Health Study, prenatal exposure to TCDD did not affect the second to fourth digit length ratio, reverse learning/set shifting, memory, attention/impulsivity, and nonverbal intelligence in children born to mothers exposed to TCDD [160,161]. Conversely, the estimated TCDD exposure levels in mothers were negatively correlated with body mass index of their female, but not male, children, whereas they were positively associated with the occurrence of metabolic syndrome in male children [162]. Moreover, the estimated TCDD levels in mothers were correlated with reduced free triiodothyronine levels in their children [163]. Children born to women with Yucheng exhibited abnormal brain plasticity as assessed via neuroimaging using functional magnetic resonance imaging [164]. Prenatal exposure to PCBs and PCDFs in the F1 generation in a Yucheng cohort also induced a low frequency of hearing loss in the right ear, and this finding was associated with the estimated maternal concentrations of 2,3,4,7,8-PeCDF at the time of birth [165].

### 4.10. Perinatal Abnormalities and Impact on Future Generations in Animal Models

Animal models have revealed various effects of perinatal dioxin exposure on future generations. Exposure to high concentrations of dioxin induced oncogenesis and systemic toxicity [166], whereas low-level exposure also influenced the next generation [167]. Prenatal exposure to low-dose dioxins induced genital malformation and sexual immaturity in rat offspring [168,169]. Sex differences in the brain are established during a developmentally sensitive perinatal period [170]. Masculinization of the brain requires testosterone-derived E2 in this sensitive period to induce male-pattern organization (maturation) of the brain, as reviewed elsewhere [170]. Perinatal exposure to dioxins can feasibly disrupt the masculinization process of the brain [171,172].

High-dose exposure to TCDD (roughly 15 μg/kg) reduced testicular testosterone production, plasma testosterone levels, and fertility in adult rats [167]. However, perinatal (in utero) exposure to low-dose TCDD (1 μg/kg) on gestational day 15 also reduced testicular and ovarian steroidogenic protein levels in male and female rat fetuses, respectively [169,173]. The reduced levels of luteinizing hormone (LH) in the pituitary gland are attributable to reproductive tissue dysfunction [174,175]. The TCDD-mediated downregulation of LH and subsequent dysfunction of reproductive organs are associated with impaired sexual behavior in adulthood in the affected individuals [175]. Sexual immaturity in both male and female rat fetuses is restored by the administration of equine chorionic gonadotropin (eCG), which exerts rat LH-like activity, in a dose-dependent manner [175,176].

The pathogenic mechanisms of TCDD-mediated LH downregulation in the pituitary gland in the hormone-sensitive perinatal period are not fully understood. Because the induction of histone deacetylases 1, 5, and 7 occurs simultaneously with the downregulation of LH, epigenetic modulation by TCDD is suspected [177]. Signs of male sexual immaturity, such as attenuated mounting behavior, induced by perinatal exposure to TCDD are also associated with reduced production of gonadotropin-releasing hormone (GnRH), and supplementation of GnRH can normalize male mounting behavior [178].

C57BL6 mice are more sensitive to dioxin toxicity than DBA mice. In parallel, higher doses of TCDD are required to induce similar decreases of pituitary LH levels in DBA fetuses than in their C57BL6 counterparts [179]. The toxicity of TCDD (TEF = 1.0) was higher than that of 2,3,4,7,8-PeCDF (TEF = 0.3) as assessed by their ability to induce CYP1A1 gene transcription [4,5]. However, the ability of 2,3,4,7,8-PeCDF to reduce pituitary LH levels is much weaker than that of TCDD, and the relative potency of the former is only 0.02-fold of that of the latter, suggesting that the biological effects of TCDD were extremely serious particularly on pituitary LH production, when administered in the hormone-sensitive perinatal period [176]. The biological effects of dioxins on the next generation should be carefully monitored in each human disaster. Because the Seveso accident was caused by TCDD exposure, special attention should be paid to this substance.

The dioxin–AHR axis induces the production of xenobiotic-metabolizing enzymes such as CYP1A1 and CYP1A2. Yusho and other studies revealed that CYP1A2 binds 2,3,4,7,8-PeCDF with high affinity and retains 2,3,4,7,8-PeCDF in the liver for a long period [180,181]. In line with this notion, the hepatic retention of dioxin was not found in *Cyp1a2*-deficient mice [182]. In addition, it is known that PCDFs are more likely to remain in the mother than in the fetus, and their transplacental transfer is less than that of PCDDs [183]. Research in *Ahr*-deficient rat proved that the dioxin–AHR axis induces hepatotoxicity by increasing leukotriene B4 synthesis and the subsequent recruitment of neutrophils [184]. A significant decrease of pituitary LH levels was observed in *Ahr*-deficient rat fetuses [185]. In addition, *Ahr*-deficient rats exhibited similar sexual immaturity as those exposed to TCDD in the hormone-sensitive perinatal period [185]. These studies strongly suggest an essential regulatory role of AHR in pituitary LH production and subsequent steroidogenesis in reproductive organs [185]. Exaggerated activation of AHR by TCDD may disrupt the homeostatic regulation of pituitary LH production to a similar extent as AHR deficiency.

As mentioned previously, the TCDD-mediated reduction of pituitary LH levels is restored by the direct injection of eCG to TCDD-exposed fetuses [174,175]. However, this approach is too invasive when we consider the therapeutic approach. We also found that decreased LH levels in the pituitary glands are associated with reduced levels of α-lipoic acid in the hypothalamus of fetuses born from dams exposed to TCDD in the hormone-sensitive perinatal period [173]. Notably, the maternal administration of α-lipoic acid can reverse the TCDD-mediated reduction of pituitary LH levels and testicular testosterone levels in the fetus [173].

In addition to sexual immaturity, perinatal exposure to dioxins in rat dams is associated with various fetal and childhood abnormalities including short stature, low birth weight [186,187], learning and memory dysfunction [188,189], and deteriorated social activity [190]. Low birth weight [139] and cognitive dysfunction [191] have been also reported in the second generation of patients with Yusho and Yucheng.

Low-dose exposure to TCDD (1 μg/kg) on gestational day 15 can induce fetal growth delay [192]. This effect is attributable to the downregulation of growth hormone and thyroid hormone in the affected fetus [192,193]. Decreased growth hormone levels, especially in male fetuses, are probably induced by the reduction of corticosterone level in mothers [192]. Therefore, supplying corticosterone to TCDD-exposed mothers reverses the TCDD-induced reduction in pup weight and decreased levels of growth hormone [192].

Decreased thyroxin production is transiently observed on gestational day 21 and postnatal day 21 in fetuses born to mothers exposed to TCDD on gestational day 15 [193]. This finding is likely related to the transient reduction of thyroxine levels on gestational days 20–21 in TCDD-exposed mothers [193]. As several uridine 5′-diphospho (UDP)-glucuronosyltransferases are upregulated in mothers on gestational day 20, the reduction of maternal thyroxine levels may be attributable to the upregulation of UDP-glucuronosyltransferases, which metabolize thyroxine [192]. Similarly, the reduction of fetal thyroxine levels is likely to be induced by the increased production of UDP-glucuronosyltransferases [192].

Another problem in the mother–child relationship induced by perinatal dioxin exposure is the nursing negligence of TCDD-exposed dams in rats [194]. TCDD-induced maternal nursing negligence is attributable to the decreased levels of maternal prolactin, the production of which is AHR-dependent [194]. Nursing negligence of mothers is responsible for learning disabilities in offspring [194]. Supplementation of prolactin to mothers reverses maternal nursing negligence and learning disabilities in offspring [194]. The upregulation of transforming growth factor-β1 appears to contribute to the decreased production of prolactin in rat mothers exposed to TCDD [194].

Careful consideration is necessary to interpret the experimental findings of animal models because rodents are more sensitive to dioxin-induced toxicity than humans [176,179,195]. However, recent human birth cohort studies described noticeable effects of dioxin on sexual function [158,196,197,198]. Patients with Yusho and Yucheng were exposed to extremely high concentrations of dioxins, and thus, the transfer of dioxins through the placenta and breastfeeding may potentially affect the next generation directly or epigenetically. In a Yucheng cohort, epigenetic differential changes were found in the DNA methylation of several genes including aryl hydrocarbon receptor repressor and CYP1A1 genes in the second generation [199].

## 5. High Blood Concentrations of Dioxins Are Sustained in Patients with Yusho

The tolerable daily intake (TDI) of dioxins is currently 4 pg-TEQ/kg/day [15]. The daily intake of dioxin-like congeners was estimated to be as high as 10 μg-TEQ/kg/day in patients with Yusho in 1968 [15]. Considering that the 50% lethal dose in guinea pigs is 1 μg-TEQ/kg/day, these patients were exposed to extremely high concentrations of PCDFs and PCBs [15]. The blood concentrations of dioxins were estimated to exceed 60,000 pg-TEQ/g-lipids in highly exposed patients with Yusho [15]. In 2004, Ukrainian president Viktor Yushchenko was subject to an attempted assassination using pure TCDD. He experienced similar symptoms as patients with Yusho, and his blood concentrations of TCDD reached 108,000 pg/g-lipids [200,201]. Dioxins and PCBs in the body are gradually and slowly metabolized by CYP1A1, CYP1A2, and CYP1B1 via an AHR-mediated process. However, the affinity of xenobiotic enzymes for these different congeners varies [202]. They are also slowly excreted from the body via sebum, sputum, and feces.

Notably, blood concentrations of 2,3,4,7,8-PeCDF in patients with Yusho remained elevated in a recent analysis compared with those in normal healthy controls. The mean, standard deviation, maximum, and minimum concentrations of 2,3,4,7,8-PeCDF were 192, 252, 1890, and 3.1 pg/g lipids, respectively, in patients with Yusho (N = 279), whereas those of normal healthy controls (N = 127) were 17, 6.6, 37, and 5 pg/g lipids, respectively [203]. Many patients with Yusho still exhibit tenfold higher blood concentrations of 2,3,4,7,8-PeCDF than healthy controls [203]. The blood concentrations of hexachlorinated biphenyl (hexaCB)-156, hexaCB-157, heptaCB-181, and heptaCB-189 were also 3.4-, 3.8-, 3.9-, and 3.8-fold higher than those of healthy controls, respectively [204]. The blood contaminant patterns in healthy controls were 45.1% for dioxin-like PCBs, 33.3% for PCDDs, 25.7% for non-ortho PCBs, 21.6% for PCDFs, and 19.4% for mono-ortho PCBs Conversely, the findings in patients with Yusho were 64.8% for PCDFs, 21.3% for dioxin-like PCBs, 15.5% for mono-ortho PCBs, 12.1% for PCDDs, and 7.6% for non-ortho PCBs, suggesting that the PCDF-dominant pattern is retained in patients with Yusho [203].

At 15 years after onset, the half-lives of most dioxins are estimated as 2.4–4.1 years in patients with Yusho [15]. However, recent studies demonstrated a prolongation of the half-life of 2,3,4,7,8-PeCDF in patients with Yusho [205,206,207]. The individual half-life of blood 2,3,4,7,8-PeCDF was diverse in 327 patients with Yusho in 2001–2006, including a group exhibiting a half-life of less than seven years and one displaying a half-life exceeding 40 years [205]. A later analysis conducted in 2002–2016 revealed an increased number of subjects with half-lives exceeding 40 years [206]. To exclude the possible involvement of daily dietary intake, the half-life of 2,3,4,7,8-PeCDF in blood was compared with that of octachlorodibenzodioxin, which is a rich contaminant in the diet and environment. The result confirmed that the prolongation of the half-life of 2,3,4,7,8-PeCDF in blood in patients with Yusho was independent of environmental factors [207]. Patients with Yusho and Meibomian gland hypersecretion or chloracne tended to exhibit short half-lives of 2,3,4,7,8-PeCDF in blood, suggesting a possibility that they may actively excrete 2,3,4,7,8-PeCDF from the Meibomian or sebaceous glands [208].

## 6. Therapeutic Approach in Yusho

Many patients with Yusho have persistently high blood concentrations of 2,3,4,7,8-PeCDF. Its blood concentrations are positively correlated with general malaise, respiratory symptoms (sputum and coughs), peripheral nerve symptoms (paresthesia and numbness of extremities), and skin symptoms (chloracne) [10,11,70,71,72,73]. Therefore, it is extremely important to identify a therapeutic approach to increase the excretion or minimize the toxicity of dioxins.

### 6.1. Trials to Enhance Dioxin Excretion

Autopsy results in rats, monkeys, and patients with Yusho illustrated that orally administered PCBs and 2,3,4,7,8-PeCDF accumulated in adipose tissue and the liver [64,209,210,211,212]. PCBs were preferentially detected in adipose tissue relative to the liver, whereas 2,3,4,7,8-PeCDF preferentially accumulated in the liver [212]. In addition, these congeners were not easily metabolized [213]. The liver-dominant accumulation of 2,3,4,7,8-PeCDF was likely to be involved in hypertriglyceridemia [214] and higher mortality for liver cancer [122].

Although PCBs are excreted in bile from the liver, the excretion of 2,3,4,7,8-PeCDF is not observed in bile [215,216,217]. Small amounts of 2,3,4,7,8-PeCDF are directly excreted from blood into feces through the small intestine, but the majority of excreted 2,3,4,7,8-PeCDF was reabsorbed in the digestive tract [215,216,217]. Thus, nonabsorbable lipidlike materials and drugs may trap PCBs and 2,3,4,7,8-PeCDF and enhance their fecal excretion. In a rat model, the oral administration of liquid paraffin [218], squalane [216,219], activated charcoal [217], or the anion exchange resin cholestyramine [217] was demonstrated to increase the fecal excretion of dioxins. Decreased blood concentrations of PCBs were observed in human subjects following the oral administration of the anion exchange resin colestimide [220]. Reduced blood concentrations of TCDD and PCBs were also reported following the oral administration of olestra, a dietary nonabsorbable lipid substitute [221,222,223]. However, the ability of these agents to increase the excretion of dioxins was not potent, and large interindividual variation was noted. In addition, the oral administration of these artificial materials is difficult to maintain over the long term.

Six patients with Yusho took cholestyramine for six months, and increased fecal excretion of PCDF was found in two patients [224]. Because dietary fibers increased the cholestyramine-induced fecal excretion of dioxins in rats [225,226], four patients with Yusho and eight patients with Yucheng were treated with cholestyramine and rice bran dietary fibers. This treatment enhanced the fecal excretion of PCBs and 2,3,4,7,8-PeCDF; however, it did not improve the patients’ symptoms [227,228,229]. A crossover clinical trial involving six months of colestimide administration enrolled 36 patients with Yusho. Twenty-six patients completed the trial, whereas nine withdrew because of constipation and abdominal pain. Post-treatment blood examination was not performed in one patient. Colestimide treatment did not reduce the blood concentrations of 2,3,4,7,8-PeCDF [230].

### 6.2. Agents to Minimize Dioxin Toxicity

Dioxins strongly activate AHR, upregulate CYP1A1, and stimulate excessive ROS generation [20,24,32]. Dioxin-mediated oxidative stress is involved in dioxin toxicity [20,24,32]. To neutralize ROS, the activation of NRF2 and subsequent induction of antioxidative enzymes are effective [39,40]. An agent that both inhibits AHR and stimulates NRF2 activity may be useful for the treatment of Yusho (Figure 5) [48]. Various phytochemicals exhibit such effects [21]. Kampo herbal medicines are rich sources of phytochemicals that potentially exhibit the desired dual (AHR inhibition and NRF2 stimulation) activity.

After screening Kampo remedies, we found that Keishi (cinnamon) and its active ingredient cinnamaldehyde have dual activity [231]. The commercially available Kampo formula Keishi-bukuryo-gan^®^ contains Keishi. Keishi-bukuryo-gan^®^ also displayed dual activity and exerted potent antioxidative activity (Figure 5) [231]. A clinical trial was conducted to treat patients with Yusho with Keishi-bukuryo-gan^®^ [19]. After three months of oral treatment, Keishi-bukuryo-gan^®^ significantly improved general malaise, sputum and cough, and chloracne in 42 patients with Yusho [19]. It also improved the physical, mental, and social quality of life (QOL) of patients with Yusho as assessed using the MOS 36-Item Short Form Health Survey^®^ (SF-36^®^) QOL questionnaire [19].

We also demonstrate that Wogon (*Scutellaria baicalensis* root) and its active ingredient baicalein inhibit AHR and stimulate NRF2 [25]. Baicalein interferes with the cytoplasmic-to-nuclear translocation of AHR by inhibiting the phosphorylation of c-Src [25]. Among various Wogon-containing Kampo formulae, Oren-gedoku-to^®^ exhibits dual activity and exerts potent antioxidative effects (Figure 5) [25]. 

In addition, a clinical trial treated 25 patients with Yusho with the Kampo formula Hochu-ekki-to^®^, Keigai-rengyo-to^®^, Gosha-jinki-gan, or Bakumondo-to^®^ [232]. In the study, Bakumondo-to^®^ significantly improved sputum and cough in patients with Yusho [232]. The compound in Bakumondo-to responsible for its effects on Yusho symptoms is unknown. Because Keishi-bukuryo-gan^®^, Oren-gedoku-to^®^, and Bakumondo-to^®^ are all commercially available Kampo formulae, these remedies are used in the treatment of Yusho in Japan. 

Many vegetables contain healthy antioxidative phytochemicals. The Ministry of Health, Labour and Welfare of Japan recommends the consumption of more than 350 g of vegetables per day. We also recommend that patients with Yusho should eat vegetables. In this context, we recently found that perillaldehyde contained in *Perilla frutescens* also exhibits potent antioxidative capacity via AHR-inhibiting and NRF2-activating effects [233].

## 7. Conclusions

Yusho first emerged in 1968, and 2,3,4,7,8-PeCDF is the major causative congener among various dioxins. The blood levels of dioxins in patients with Yusho were expected to gradually normalize. However, this has not occurred. The half-lives of dioxins are prolonged, and extremely high blood concentrations of dioxins are still detected in many patients with Yusho more than 50 years after onset. These patients thus experience continuous exposure to high levels of dioxins throughout their lives. Most of them exhibit various symptoms such as general malaise, chloracne and scars, cough and sputum, and paresthesia. We did not address other rare dioxin-related symptoms such as porphyrinopathy, hirsutism and palmar hyperkeratosis in this article [234,235,236]. 

It has not been possible to remove contaminating dioxins from blood to date. However, research on AHR is rapidly progressing. We can inhibit dioxin-mediated AHR activation and subsequent oxidative stress using various antioxidative phytochemicals. This research concept may be applicable to other disease fields and may facilitate the development of new drugs.

## Figures and Tables

**Figure 1 ijms-22-00708-f001:**
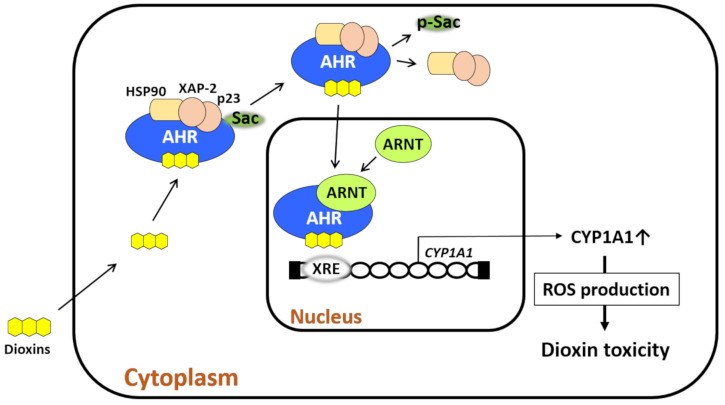
Dioxin-mediated activation of aryl hydrocarbon receptor (AHR) induces oxidative stress. AHR resides in the cytoplasm with chaperone proteins (heat shock protein 90 [HSP90], X-associated protein 2 [XAP-2], and p23) and c-Src. Upon ligation by dioxin, AHR leaves this chaperone complex upon c-Src phosphorylation. AHR then translocates from the cytoplasm to the nucleus, in which the dioxin–AHR complex couples with AHR nuclear translocator (ARNT). The dioxin–AHR–ARNT complex binds the xenobiotic response elements (XREs) of target genes such as cytochrome P450 1A1 (CYP1A1), CYP1A2, and CYP1B1. CYP1A1 metabolizes the ligand and induces reactive oxygen species (ROS) generation during the metabolic process. Dioxins are chemically stable and tolerant to CYP1A1 activity, and thus, they induce high levels of ROS production. Dioxin-induced oxidative stress characterizes dioxin-mediated toxicity.

**Figure 2 ijms-22-00708-f002:**
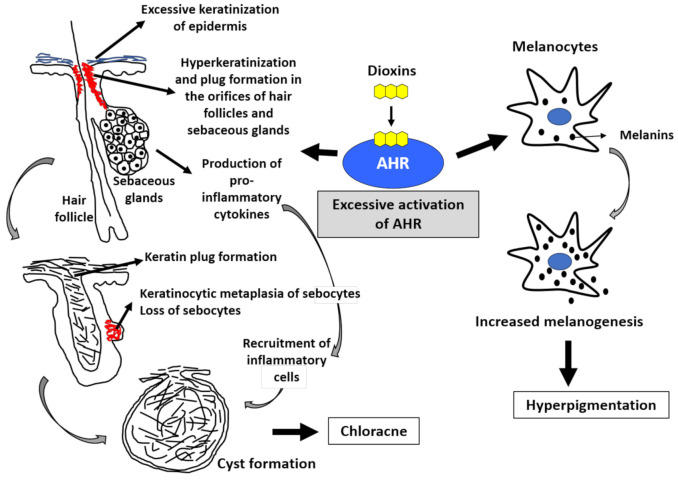
Excessive activation of aryl hydrocarbon receptor (AHR) by dioxins induces chloracne and hyperpigmentation. The activation of AHR induces keratinization (upregulated epidermal differentiation). Excessive activation of AHR by dioxins induces epidermal hyperkeratinization. It also induces hyperkeratinization and keratin plug formation in the orifices of hair follicles and sebaceous glands. Most importantly, excessive AHR activation induces keratinocytic metaplasia of sebocytes, impairs sebum production, and finally results in the loss of sebocytes. Hair follicles eventually form cystic appearance. Activation of AHR by dioxins also induces the production of proinflammatory cytokines and accelerates the recruitment of inflammatory cell infiltration. These epithelial alterations and inflammatory processes lead to the development of chloracne. Melanocytes produce melanins. Excessive AHR activation by dioxins upregulates melanogenesis, and hyperpigmentation occurs.

**Figure 3 ijms-22-00708-f003:**
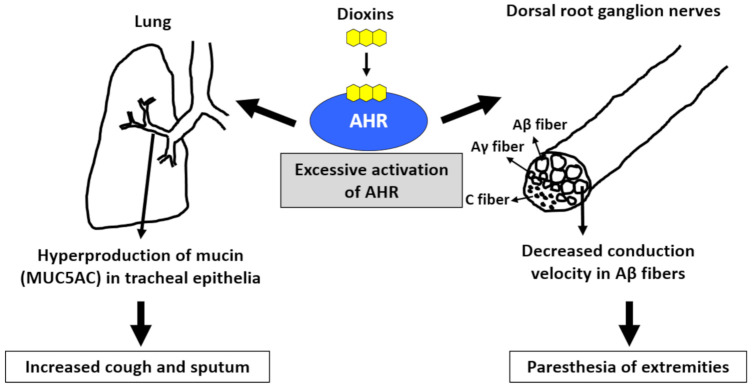
Excessive activation of aryl hydrocarbon receptor (AHR) by dioxins induces cough and sputum as well as paresthesia of the extremities. Activation of AHR increases the production of mucin (MUC5AC) in tracheal epithelial cells, which may cause increased cough and sputum. Activation of AHR ameliorates the conduction velocity of Aβ fibers in the dorsal root ganglion nerves. This may cause paresthesia of the extremities.

**Figure 4 ijms-22-00708-f004:**
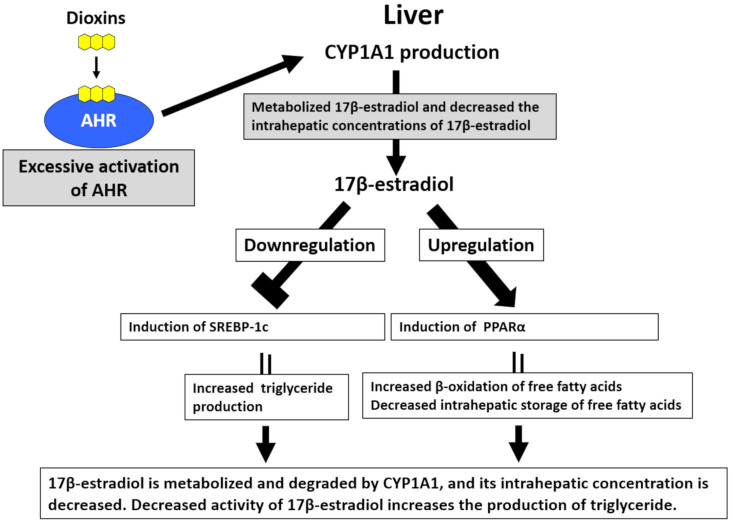
Excessive activation of aryl hydrocarbon receptor (AHR) by dioxins induces hypertriglyceridemia. In hepatocytes, sterol regulatory element-binding protein-1c (SREBP-1c) increases triglyceride production, and peroxisome proliferator-activated receptor-α (PPARα) upregulates the β-oxidation of free fatty acids and reduces the intrahepatic storage of free fatty acids. 17β-estradiol inhibits the activity of SREBP-1c and enhances PPARα activity, subsequently reducing triglyceride production. Hepatocytes abundantly express AHR. Upon the activation of AHR by dioxins, the expression of cytochrome P450 1A1 (CYP1A1) is augmented. CYP1A1 metabolizes 17β-estradiol, thereby decreasing its intrahepatic concentrations and enhancing triglyceride production.

**Figure 5 ijms-22-00708-f005:**
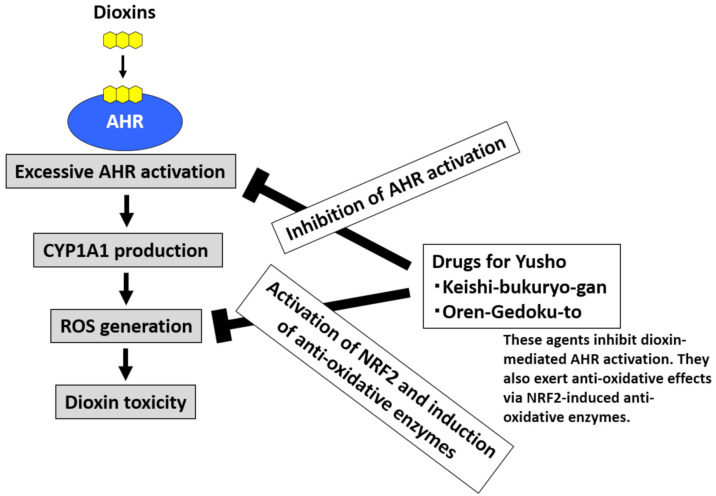
Development of drugs for Yusho. Dioxins activate aryl hydrocarbon receptor (AHR) and induce cytochrome P450 1A1 (CYP1A1) production. CYP1A1 enhances the production of reactive oxygen species (ROS), which are responsible for the toxic effects of dioxins. Kampo herbal medicines contain many phytochemicals that can inhibit AHR activation and stimulate the antioxidative system via nuclear factor-erythroid 2-related factor-2 (NRF2) activation. The commercially available Kampo formulae Keishi-bukuryo-gan and Oren-gedoku-to both inhibit AHR and stimulate NRF2 activity. These remedies are potentially useful for the treatment of Yusho and other health hazards associated with dioxin exposure.

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
