# Peer review of "Aryl Hydrocarbon Receptor and Dioxin-Related Health Hazards—Lessons from Yusho"

_ijms, 2021, doi:10.3390/ijms22020708_

Round 1

Reviewer 1 Report

This interesting review manuscript describes the mechanisms of the toxicity of Dioxin and its metabolites in humans and animals in the Yusho disease. The authors explained in detailed the pathogenic mechanisms of the toxicity of Dioxin that have been represented as cartoon figures.   Dioxins exert their toxicity via aryl hydrocarbon receptor (AHR) through the generation of reactive oxygen species (ROS). Furthermore, the authors explained the potential therapeutic use of herbal drugs that are targeting AHR and ROS in patients with Yusho.

However, it would be better if the authors can address the followings

1- The antioxidant network in Yusho patients.

2- Is there any evidence of the epigenetic alterations in Humans

Author Response

Reply to the Reviewer 1

This interesting review manuscript describes the mechanisms of the toxicity of Dioxin and its metabolites in humans and animals in the Yusho disease. The authors explained in detailed the pathogenic mechanisms of the toxicity of Dioxin that have been represented as cartoon figures. Dioxins exert their toxicity via aryl hydrocarbon receptor (AHR) through the generation of reactive oxygen species (ROS). Furthermore, the authors explained the potential therapeutic use of herbal drugs that are targeting AHR and ROS in patients with Yusho.

→ Thank you very much for your very encouraging comment.

However, it would be better if the authors can address the followings

1- The antioxidant network in Yusho patients.

→ Thank you very much for your very critical comment. Unfortunately, we did not address the antioxidant network in Yusho patients. We would like to check it in the future project.

2- Is there any evidence of the epigenetic alterations in Humans

→ Thank you very much for your important comment. We did not yet examine the epigenetic alteration in Yusho patients. However, in Yucheng cohort in Taiwan, they found an epigenetic differential changes in the DNA methylation of several genes including aryl hydrocarbon receptor repressor and CYP1A1 genes in the second generation [199]. This was mentioned in line 526-528 in the revised manuscript.

Thank you very much for your very helpful comments.

We hope the revised article is suitable for publication in IJMS.

Reviewer 2 Report

The review paper of Furue et al represents an outstanding overview of AHR related effects seen in the Yusho cohort and related studies. The paper is very well written und easily understandable. It is also interesting for a broad audience. A clear overview of the pathogenesis is given and sometimes illustrated in Figures

Some minor comments:

Keywords: The word "PCB" is missing

Introduction: The exposure was to PCDD/Fs and PCBs. Were the effects all AhR related? What about the non-AhR effects of (ndl)PCBs?

Good overview of the Yusho disease is given. Besides the mentioned symptoms, how many yusho victims deceased shortly after the incident?

Methods: A great overview of previous work on the Yusho cohort and relevant related papers are given. However it is not clear how the selection of used papers in the review is made. A small chapter ‘materials and methods’ with an overview of this selection and used databases including search therms might improve the paper.

Chapter 2: Page 3: “Although tryptophan photoproducts”. This sentence might benefit from further explanation considering its role in the AHR activation.

Chapter 3.1:

Great overview of chloracne and hyperpigmentation, the main cutaneous effects. It might be interesting to also mention shortly other cutaneous effects seen, like porphirinopathy, hirsutismus, palmar hyperhidrosis and hyperkeratosis.

Figure 2 and 3: Very useful to understand pathogenesis, however some of the drawings could be improved.

Chapter 3.8: Mortality rates on long-term are well described. What about the direct mortality in the most highly affected victims?

Fig 2 gives a good overview of pathogenesis in the skin. Production of pro-inflammatory cytokins as well as ROS production might lead to higher incidence of cutaneous malignancies. What about cutaneous malignancies in the Yusho cohort and other studies?

Author Response

Reply to the Reviewer 2

The review paper of Furue et al represents an outstanding overview of AHR related effects seen in the Yusho cohort and related studies. The paper is very well written und easily understandable. It is also interesting for a broad audience. A clear overview of the pathogenesis is given and sometimes illustrated in Figures

→ Thank you so much for your encouraging comment.

Some minor comments:

Keywords: The word "PCB" is missing

→ We added "PCB" in the Keywords. Thank you very much.

Introduction: The exposure was to PCDD/Fs and PCBs. Were the effects all AhR related? What about the non-AhR effects of (ndl)PCBs?

→ Thank you so much for your helpful comment. According to your comment, we added the following sentences in the revised article.

Line 83-84

“Some AHR-binding pollutants also exert endocrine-dsrupting activity [26].”

Good overview of the Yusho disease is given. Besides the mentioned symptoms, how many yusho victims deceased shortly after the incident?

→ Thank you very much for your critical comment. In 1968 and 1969, patients with skin changes (chloracne and hyperpigmentation) had been registered as Yusho. Their early medical records were not preserved. Therefore, further statistical analysis was not precisely performed in early studies. We are sorry that we cannot respond to your comment.

Methods: A great overview of previous work on the Yusho cohort and relevant related papers are given. However it is not clear how the selection of used papers in the review is made. A small chapter ‘materials and methods’ with an overview of this selection and used databases including search therms might improve the paper.

→ Thank you very much for your helpful comment. We agree with your suggestion. According to your comment, we added in the following sentences in the revised article.

Line 60-69

“2. Methods

Using free text and key words, we searched PubMed (https://www.ncbi. nlm.nih.gov/pubmed/advanced), Google Scholar (https://scholar.google.co.jp/schhp?hl=ja) and Japan Medical Abstracts Society (https://search.jamas.or.jp/) databases. Studies were limited to those published from January 1968 to November 2020 with no language restrictions. We included studies of Yusho, Yucheng, Seveso, dioxin, PCB, AHR, ROS, chloracne, pigmentation, lung, cough, sputum, nerve, paresthesia, triglyceride, sleep, circadian clock, immune response, Th22, mortality, carcinogenesis, perinatal, infant, offspring, breast milk, sex ratio, blood concentration, halflife, NRF2, treatment, and/or phytochemicals.”

Chapter 2: Page 3: “Although tryptophan photoproducts”. This sentence might benefit from further explanation considering its role in the AHR activation.

→ Thank you so much for your recommendation. According to your comment, we added the following sentences in the revised article.

Line 97-99

“One of the well-examined endogenous ligands is tryptophan photoproduct 6-formylindolo[3,2-b]carbazole (FICZ) which is induced by ultraviolet ray irradiation [29]. FICZ is a high-affinity ligand for AHR [29].”

Chapter 3.1:

Great overview of chloracne and hyperpigmentation, the main cutaneous effects. It might be interesting to also mention shortly other cutaneous effects seen, like porphirinopathy, hirsutismus, palmar hyperhidrosis and hyperkeratosis.

→ Thank you so much for your helpful comments. According to your comments, we added the following sentence in the revised article adding 3 references.

Line 661-662

“We did not address other rare dioxin-related symptoms such as porphyrinopathy, hirsutism and palmar hyperkeratosis in this article [234-236].”

Figure 2 and 3: Very useful to understand pathogenesis, however some of the drawings could be improved.

→ Thank you for your comment. We agree with you. According to your comment, we amended the Fig.2 and Fig. 3.

Chapter 3.8: Mortality rates on long-term are well described. What about the direct mortality in the most highly affected victims?

→ Thank you very much for your critical comment. However, there was no precise data on the acute (or direct) death in Yusho study.

Fig 2 gives a good overview of pathogenesis in the skin. Production of pro-inflammatory cytokins as well as ROS production might lead to higher incidence of cutaneous malignancies. What about cutaneous malignancies in the Yusho cohort and other studies?

→ Thank you very much for your critical comment. Your question is very reasonable and we also thoroughly checked this point through annual medical check-up. However, there was no evidence that dioxin poisoning increase cutaneous malignancy.

One of the plausible explanation is that AHR activation accelerates the keratinization (differentiation) process. Therefore, oncogenic clones may be released out from epidermis in very early stage. However, this is just a hypothesis.

Thank you so much again for your important comments.

We hope the revised manuscript is now suitable for publication in IJMS.
